# Socioeconomic and health impacts of fall armyworm in Ethiopia

Zewdu Abro[1], Emily Kimathi[2], Hugo De Groote[3], Tadele Tefera[1], Subramanian Sevgan[2], Saliou Niassy[2], Menale Kassie[2]*

1 International Centre of Insect Physiology and Ecology (*icipe*), Addis Ababa, Ethiopia, 2 International Centre of Insect Physiology and Ecology (*icipe*), Nairobi, Kenya, 3 International Maize and Wheat Improvement Center (CIMMYT), Nairobi, Kenya

* mkassie@icipe.org

**Data Availability Statement:** The community survey data are available in the Supporting information of this submission. The household survey are available upon request of the Central Statistics Agency of Ethiopia (https://www.

## Abstract

Since 2016, fall armyworm (FAW) has threatened sub-Saharan 'Africa's fragile food systems and economic performance. Yet, there is limited evidence on this transboundary pest's economic and food security impacts in the region. Additionally, the health and environmental consequences of the insecticides being used to control FAW have not been studied. This paper presents evidence on the impacts of FAW on maize production, food security, and human and environmental health. We use a combination of an agroecology-based community survey and nationally representative data from an agricultural household survey to achieve our objectives. The results indicate that the pest causes an average annual loss of 36% in maize production, reducing 0.67 million tonnes of maize (0.225 million tonnes per year) between 2017 and 2019. The total economic loss is US$ 200 million, or 0.08% of the gross domestic product. The lost production could have met the per capita maize consumption of 4 million people. We also find that insecticides to control FAW have more significant toxic effects on the environment than on humans. This paper highlights governments and development partners need to invest in sustainable FAW control strategies to reduce maize production loss, improve food security, and protect human and environmental health.

## Introduction

Maize is a staple food for more than 300 million Africans [1,2]. Despite the importance of maize, its production is constrained by several biotic and abiotic factors that contribute to sub-Saharan 'Africa's (SSA) pervasive food insecurity. For a long time, stemborers and Striga weed were the main maize pests in SSA, a combination known to cause complete maize production failure [3]. The recent invasion (since 2016) of maize by fall armyworm (FAW), *Spodoptera frugiperda*, hereafter referred to as FAW, has exacerbated the already fragile food systems and food security in the region [4–8]. Farm-level estimates in some SSA countries showed that FAW causes maize production losses of between 11% and 67% [6,9–14].

statsethiopia.gov.et/); The names of the datasets are Agriculture Sample Survey 2017/2018 (2010 E. C.), Agriculture Sample Survey 2018/2019 (2011 E. C.), and Agriculture Sample Survey 2019/2020 (2012 E.C.).

**Funding:** This study was supported by the USAID Feed the Future IPM Innovation Lab, Virginia Tech (Grant No. AID-OAA-L-15-00001); the Norwegian Agency for Development Cooperation (NORAD, Grant No. RAF-3058 KEN-18/0005); and the European Commission (Grant No. DCI-FOOD/ 2018/402-634). We also acknowledge the International Centre of Insect Physiology and Ecology (icipe) core support provided by the Foreign, Commonwealth and Development Office (FCDO), UK; the Swedish International Development Cooperation Agency (Sida); the Swiss Agency for Development and Cooperation (SDC); Germany's Federal Ministry for Economic Cooperation and Development (BMZ); the Federal Democratic Republic of Ethiopia; and the Kenyan Government. The funders had no role in study design, data collection and analysis, decision to publish, or preparation of the manuscript.

**Competing interests:** The authors have declared that no competing interests exist.

Infestation by invasive transboundary pests such as FAW also causes additional costs due to insecticide use and labor to control the pest [10,15–17]. The application of insecticides is the primary FAW control strategy in SSA countries [18]. This inevitably has impacts beyond abating maize production losses; insecticide pollution can adversely affect the environment, biodiversity, and health of the producers and consumers [12,19–22]. Furthermore, FAW invasions can affect trade, income, and food consumption due to reductions in maize supply. FAW invasions can also increase health expenditure arising from exposure to insecticides and affect the performance of businesses along the maize value chain, such as maize input suppliers and contributors to the livestock feed sector [23–25]. Unless effective control strategies are implemented, the pest will continue to cause massive destruction to maize and affect the livelihoods of millions of people in SSA. Implementing such control strategies requires updates on the current impact of FAW on the economy, food security, and health (human and environmental).

Despite FAW's economic importance, there are limited studies on its impact on production, the cost of control, including insecticides, and the unintended negative consequences of insecticide use on human and environmental health. Using survey data from Ghana and Zambia, Day, Rwomushana and colleagues extrapolated production losses due to FAW for twelve SSA countries [12,26]. Country-specific studies are crucial because the effects of FAW vary across and within countries due to differences in agro-ecology, farming practices, and farm and farmer characteristics. De Groote and colleagues showed that losses caused by FAW vary by agro-ecological zones [6]. Many of the existing studies did not capture the large degree of agro-ecological and socioeconomic heterogeneity of smallholder farmers in SSA because the studies rely on limited geographical areas [9,10,13,27]. Many of these studies also use data collected at the early stages of FAW invasion. The real impacts of FAW infestation may take time to become evident as the infestation varies from season to season. The arrival of FAW has changed the dynamics of existing farming system constraints to maize production, leading to a new status quo [28].

Invasion by FAW has significantly increased insecticide use in most invaded regions [10,17]. The majority of studies have focused on the efficacy of insecticide for the management of FAW in the invaded regions [29,30], but the increased use of insecticides for FAW control is affecting the health of farmers. For instance, farmers have reported sickness in Ghana and Zambia after applying insecticides recommended for controlling FAW [12]. However, no systematic study has been conducted so far to document the health and environmental effects of insecticides.

In this paper, we present evidence on the economic and health cost of FAW. Particularly, we estimated the pest's effect on maize production, food security, and the effects of insecticide use on public and environmental health. The evidence will help prioritize investment in FAW management strategies that simultaneously reduce losses and maintain ecological balance. As secondary objectives, we endeavor to understand farmers' current FAW control measures, effectiveness, and support that communities receive in combating the pest. Since the accuracy of production loss estimates depends on farmers' knowledge of the pest [31], we examined farmers' awareness and knowledge about FAW.

To measure FAW's effects on maize production and achieve the secondary objectives, we combined agroecology-based community surveys using focus groups discussion (FGD) with nationally representative datasets collected by Ethiopia's Central Statistical Agency (CSA). We covered 150 villages/communities and 1,100 farmers distributed across 30 districts of maize-growing agro-ecological zones. We also collected data from 180 agricultural development agents (DAs) and their supervisors in these communities to validate the results from the community survey data. The DAs live and work in the villages, have extensive knowledge of the farming system in the study communities, and conduct campaigns and scouting exercises,

distribute insecticides and provide advisory services to farmers to manage FAW. Supervisors visit and provide technical support to DAs. We recorded yield and yield losses agreed by each member of the FGD, often after a hot debate. This can help to minimize recall bias and avoid too high or too low estimates. The datasets from the CSA's Agricultural Sample Survey showed a good picture of maize production in Ethiopia. On average, the data covers 17,833 maize-growing farmers across the country. We developed a simple arithmetic formula to quantify maize production losses at the national level. Using secondary data obtained from the Ministry of Agriculture (Ethiopia), we applied the environmental impact quotient (EIQ) approach to quantify adverse health risks and the environmental effects of insecticides used to manage FAW [32,33].

We report three key results. First, 97% and 88% of the farmers interviewed were aware of and correctly identified FAW, respectively. Knowledge of farmer's awareness of FAW is vital to estimating its effects accurately. Second, FAW has a considerable socioeconomic impact in Ethiopia that varies by agro-ecology. From 2017 to 2019, the country lost 0.67 million tonnes of maize production, worth US$ 200 million (0.08% of the Gross Domestic Product). This lost maize could have met the maize consumption requirement of 4 million food-insecure house-holds. Third, FAW has a negative spillover effect on biodiversity and the human population. In the short term, the application of insecticides to control FAW has greater potential toxic effects on the environment than on humans. It also aggravates food insecurity by killing bene-ficial insects and contaminating other essential natural resources in the long term.

Overall, our findings present a cautionary note about the impacts of FAW. Lack of appro-priate control measures against FAW combined with other production constraints can lead to high economic losses to society and monetary expenditures associated with managing this pest. Although the food security cost is not high at the household level (60 kg per year per affected farmer), the economic and biodiversity losses are high at the national level. For example, from 2017 and 2019, the country lost US$ 204 million worth of income due to maize production losses and insecticides purchases. However, if the pest persists, it can cause food security and poverty problems in the long run by reducing marketed surplus and income [10].

## Context and responses to FAW occurrence in Ethiopia

Since FAW first occurrence in West Africa in 2016, it has spread quickly throughout SSA, including Ethiopia [12,26]. In Ethiopia, FAW is a threat to over 9 million maize-growing farm households. It was first observed in 2017 [34] and is currently one of the most destructive maize pests in the country. Maize is an economically important and strategic food security crop covering 20% of cultivated land and accounting for 30% of cereal production [35]. It pro-vides the largest share of calories (22%) for most Ethiopians [36]. It is also the most productive cereal crop in the country, with an average yield of 4 tonnes/ha [35].

The sudden invasion of FAW has forced the government of Ethiopia and stakeholders to use insecticides as an emergency measure in FAW-affected maize fields. Over the study period, the Ministry of Agriculture of Ethiopia (MOA), through the Regional and District Bureaus of Agriculture, distributed 457,427 liters of insecticides, sprayed on 1.5 million ha of maize [37]. The direct cost of insecticides to the government was about US$ 4 million [37]. This does not include the insecticides that farmers purchased themselves through other means or the costs of surveillance and management, on which no data are available. While insecticides were used to reduce losses due to FAW, they have unintended consequences on human and environmental health [12,26].

## Materials and methods

### Study areas

This study covers the major maize-producing districts and agro-ecological zones of Ethiopia. We used the sampling frame prepared by the Sustainable Intensification of Maize-Legume Cropping Systems in Eastern and Southern Africa (SIMLESA) project of the International Wheat and Maize Improvement Center (CIMMYT) [38]. The SIMLESA survey was designed to represent the key maize-producing agro-ecological zones of Ethiopia. It covered 225 maize-producing villages in 39 districts in Amhara, Benishangul Gumuz, Oromia, Southern Nations and Nationalities (SNNP), and the Tigray Regional States. In our study, we cover 30 districts and 150 villages. We dropped seven districts in the Oromia Regional State due to security reasons and excluded the Benishangul Gumuz and the Tigray Regional States for logistical reasons. The three remaining Regional States (Amhara, Oromia, and SNNPR) jointly produce more than 86% of the country's maize, as reported in Table 1 [35].

The study villages and their corresponding agro-ecological zones are represented in Fig 1. The agroecological zone classifications are from the CIMMYT's maize mega-environments (MMEs). An MME is a homogenous production environment with similar agro-climatic conditions defined using rainfall and temperature [39,40]. Rainfall and temperature are key parameters that affect maize production and the biology and spread of FAW [41,42].

Among the communities we surveyed, 71 villages are classified as wet upper mid-altitudes, 48 are in the highlands, 28 are in the dry mid-altitudes, two are found in the wet lower mid-altitudes, and one is in the dry lowlands. Over the study period, nearly 96% of the maize production in the country came from three major MMEs: the wet upper mid-altitudes (45%), the highlands (39%), and the dry mid-altitudes (12%). The remainder of the country's maize production came from the wet lower mid-altitudes, wet lowlands, and dry lowlands, each contributing nearly 1% (Table 1).

### Data sources and collection

We used data from three sources: first, we used primary community survey data collected using FGD and expert opinion data collected through individual interview. These datasets collected between June and July 2020 from 150 communities. On average, seven farmers participated per FGD, making 1,100 (10% women) farmers in total. The expert opinion survey involved 180 agricultural experts, of whom 150 were development agents (DAs) who worked closely with the farmers, and 30 were DAs supervisors who worked in the districts' agriculture offices. These experts have direct knowledge and expertise on agricultural production and districts and villages' farming systems. They coordinated FAW awareness campaigns and management, conduct scouting exercises, and distributed insecticides. The DAs are also responsible for reporting production and related data to the district agricultural office, including areas affected by FAW and the numbers of farmers affected in their respective command areas. We used a structured questionnaire covering various topics, including farmers' awareness and knowledge of FAW, the percentage of farmers affected by FAW, control strategies, attainable yield, actual yield, and yield losses due to FAW. We collected data on the percentage of farmers affected by FAW in their respective villages and the yield losses due to FAW in the 2017, 2018, and 2019 production seasons.

Self-report data, especially when collected over time, might be prone to recall bias. Although we cannot entirely rule out recall bias, we developed confidence in farmers' yield and yield losses estimates for the following reasons: (1). we recorded yield and yield losses data agreed by each member of the FGD, often after a hot debate. This can help to minimize recall

**Table 1. Area under maize cultivation and production by agro-ecological zones in Ethiopia.**

| Agro-ecological zones | Cultivated land (millions of ha) | | | Production (millions of tonnes) | | |
|---|---|---|---|---|---|---|
| | **2017** | **2018** | **2019** | **2017** | **2018** | **2019** |
| Wet upper mid-altitudes | 0.85 | 0.97 | 0.85 | 3.74 | 4.27 | 3.78 |
| Wet lower mid-altitudes | 0.05 | 0.03 | 0.03 | 0.14 | 0.09 | 0.09 |
| Dry mid-altitudes | 0.34 | 0.29 | 0.34 | 1.00 | 0.88 | 1.37 |
| Wet lowlands | 0.01 | 0.03 | 0.03 | 0.04 | 0.11 | 0.13 |
| Dry lowlands | 0.04 | 0.04 | 0.02 | 0.08 | 0.05 | 0.07 |
| Highlands | 0.69 | 0.85 | 0.82 | 2.95 | 3.59 | 3.59 |
| Total | 1.98 | 2.20 | 2.08 | 7.95 | 8.98 | 9.03 |

Source: CSA's agricultural sample survey (2017–2019).

bias and avoid too high or too low estimates. This helps to minimize recall bias and avoid too high or too low estimates. A recent study [43] in Ethiopia also shows that difference in self-reported and objectively measured yield is not big. Even in some cases, farmer's reported yield is better than yield objectively measured by W-walk and Transect methods [44]; (2) because of the community-level campaigns and intensive information exchange through extension and

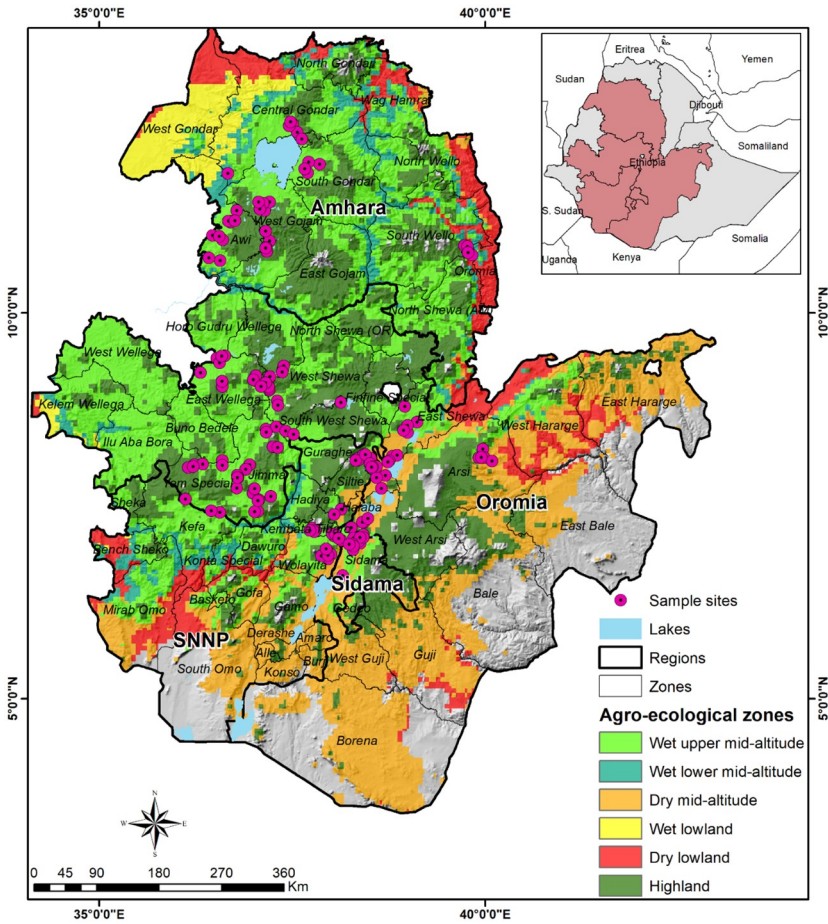

**Fig 1. The study areas and the location of sample communities within maize mega-environments.**

other local communication channels to control FAW, farmers provided attention to the pest, and its impact, which can help them provide reasonable estimates of yields and yield losses data; (3) Farmers cultivate maize on same plot year after year, and production variation across years is limited-implying recalling recent years' data might not be difficult; and (4) the data were collected after well-trained enumerators and supervisors carefully explained until the FGD participants understand the questions.

To understand farmers' FAW awareness and knowledge levels, we asked each FGD participant two questions: (1) Are you aware of FAW? and (2) Can you identify FAW from these pictures? (Fig 2). Awareness implies that the farmers have heard about FAW through fellow farmers and development agents, but they may not experience the pest in their fields. Knowledge is much more than awareness, and farmers must identify or know FAW from the pictures, and we asked question (2) if the farmers said yes to question (1). The questionnaire had an introductory statement to obtain verbal consent from respondents and agricultural experts. We have also got approval from *icipe*'s science committee.

The second dataset comes from the agricultural sample survey datasets for 2017, 2018, and 2019 main seasons. These datasets are nationally representative household survey data collected by Ethiopia's Central Statistical Agency [35]. The agency obtained verbal consent from respondents. The third datasets consist of insecticide data collected by the authors from the Ministry of Agriculture [37].

To measure the effect of FAW, we combined the community survey data with the CSA data, from which we obtained the total maize area and the number of maize-growing farmers in the country CSA datasets. We identified the agro-ecological zones for each survey community by overlaying their coordinates with the global maize mega-environments' shapefile. Because we did not have access to the farmers' coordinates in the CSA survey, we used the centroids of the CSA's survey areas to identify the key MMEs. Finally, we used region, zone, district, and MMEs as unique identifiers to combine the two datasets.

## Measuring maize production losses

Maize yield loss is the difference between attainable yield without FAW and actual yield in FAW presence [6]. However, FAW is not the only cause of yield loss. Several other factors contribute to yield loss, including abiotic factors (e.g., drought and soil fertility) and other biotic factors (e.g., diseases, stemborers, and locusts). Results may be biased if farmers are asked directly to estimate yield loss due to FAW alone without considering the potential yield loss attributable to other production constraints. To mitigate this problem, we first asked farmers to compute the actual maize yield in the community in the presence of all production constraints, including FAW. Second, we asked them to estimate the attainable yield in the absence of production constraints. Third, we asked farmers to quantify FAW's contribution and other production constraints to the yield gap, the difference between the attainable and actual yields.

We calculated the total production loss ($PL_i$) due to FAW using Eq (1) as follows:

$$PL_i = \sum_i^k A_i \times [(Y_a - Y) \times L_i] \times (N_{hi} \times F_{ai}) \tag{1}$$

where the index $i$ represents agro-ecological zones; $k$ denotes the number of agro-ecological zones; $A_i$ is the average land size (ha) devoted to maize in that zone; $Y_a$ is the attainable yield without production stresses, including FAW (tonnes/ha); and $Y$ is the actual yield in the presence of FAW and other production stresses (tonnes/ha). $L_i$ is the proportion of the average yield losses attributed to FAW (%); $N_h$ is the number of maize-growing households; and $F_{ai}$ is the proportion of farmers affected by FAW. We obtained the values for $Y_a$, $Y$, $L_i$ and $F_{ai}$ from

A1) Stem borer (*Chilo Partellus*)

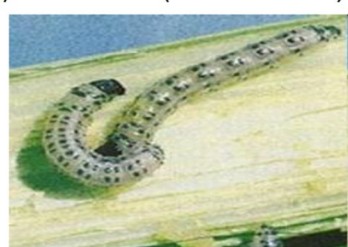

B) African armyworm

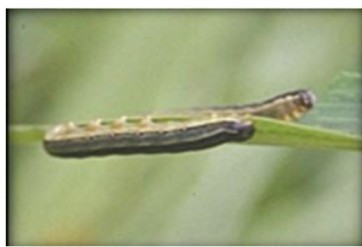

A2) Stemborer (*Busseola fusca*)

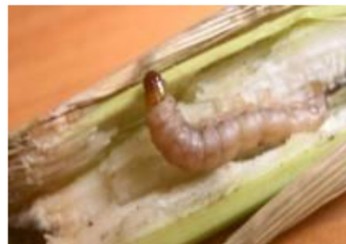

C) Fall armyworm

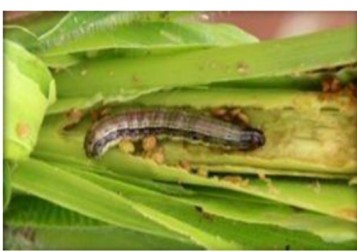

**Fig 2. Pictures of lepidopterous insect pests shown to farmers.** A) stemborers (either *Chilo partellus* (A1), or *Busseola Fusca* (A2); B) *Spodoptera exempta*; and C) *Spodoptera frugiperda, obtained from* [6] *with permission.*

the community survey data, while the values of $A_i$ and $N_{hi}$ were from the CSA datasets (Table 2).

Although farmers and government incur management costs, we focused on production losses ($PL_i$) because we did not have full management cost data. However, as discussed in the next section, we report the insecticides costs and measure the impacts of insecticides spraying on human health and the environment.

**Table 2. Attainable yield, actual yield, and average land size in Ethiopia (2017–2019).**

| Agro-ecological zones | Attainable yield (tonnes/ha)-($Y_a$) | Actual yield (tonnes/ha)-($Y$) | Yield losses due to FAW and other stresses (tonnes/ha)-($Y_a - Y$) | Average land size (ha)-($A_i$) | Number of farmers (millions) ($N_{hi}$) |
|---|---|---|---|---|---|
| | A | B | C = A-B | D | E |
| Wet upper mid-altitudes | 4.02 | 2.76 | 1.26 | 0.12 | 3.41 |
| | (0.10) | (0.08) | (0.05) | (0.08) | (0.02) |
| Wet lower mid-altitudes | 5.08 | 3.73 | 1.35 | 0.08 | 0.32 |
| | (0.90) | (0.82) | (0.15) | (0.01) | (0.01) |
| Dry mid-altitudes | 4.40 | 2.86 | 1.54 | 0.14 | 1.29 |
| | (0.13) | (0.13) | (0.07) | (0.01) | (0.01) |
| Dry lowlands | 3.10 | 2.47 | 0.63 | 0.10 | 0.33 |
| | (0.12) | (0.16) | (0.11) | (0.02) | (0.01) |
| Highlands | 4.13 | 2.88 | 1.25 | 0.11 | 3.80 |
| | (0.14) | (0.11) | (0.06) | (0.06) | (0.01) |
| Average | 4.11 | 2.82 | 1.29 | 0.12 | 9.28 |
| | (0.07) | (0.06) | (0.03) | (0.04) | (0.01) |

Note: Standard errors in parenthesis.

Sources: Columns A and B are from the community survey data; columns D and E are from the CSA's agricultural sample survey.

### Measuring the impacts of insecticides on environmental and human health

While insecticides were used to boost crop productivity, they have unintended consequences on human and environmental health. The use of insecticides poses a risk to human health, water quality, food safety, aquatic species, and beneficial insects [21,45–51]. The objective of measuring the health and environmental impacts of insecticides used to control FAW was to account for the indirect cost of the pest, regardless of who used the insecticides and their impact on individual farmers. In developing countries, insecticides spraying is often conducted manually, without adequate measures to prevent negative effects on human health and the environment [12,52].

To measure the potential risks to human health and the environment caused by insecticides used to control FAW, we used the environmental impact quotient (EIQ) [32,33]. For the empirical application of this method in the agriculture sector, see [21,48,53]. Although the EIQ uses arbitrary weights to measure the effects of the insecticides, it has been used in other studies as there is no readily available alternative to EIQ at present [21,48,54,55]. In any event, it is vital to consider the health and environmental effects [21]. We calculate the EIQ for each active insecticides ingredient as follows:

$$EIQ = [C(DT \times 5) + (DT \times P)] + [(C \times (S+P)2 \times SY) + (L)] + [(F \times R) \\ + \left(D \times \frac{S+P}{2} \times 3\right) + (Z \times P \times 3) + (B \times P \times 5)]\}/3 \tag{2}$$

where $C$ is chronic toxicity; $DT$ is dermal toxicity; $SY$ is systemicity; $F$ is fish toxicity; $L$ is leaching potential; $R$ is surface loss potential; $D$ is bird toxicity; $S$ is soil residues half-life; $Z$ is bee toxicity; $B$ is beneficial arthropod toxicity; $P$ is plant surface residues half-life. The first, the second, and the third part of the Eq (2) within the square brackets are the producer, consumer, and environmental effects of the pesticides, respectively. We used the EIQ Field Use [33]. We obtain the EIQ Field Use by multiplying the EIQ value with the active ingredient of the pesticides used and its application rate [32]. From Eq (2), the average of the producer, consumer, and environmental effects provide the EIQ. The environmental effects of insecticides include the threat to birds, bees, and other beneficial insects. It may also contaminate groundwater and soil due to the potential leaching of the insecticides [46]. Low EIQ values indicate the negative impacts of insecticides are low and vice versa. EIQ relies on published toxicology and environmental fate data [32,33].

## Results and discussion

### FAW awareness and knowledge of farmers

We found that 97% of the FGD participants were aware of FAW. Moreover, 88% of the farmers in the FGDs correctly identified FAW (Table 3), slightly more than those in Kenya (82%) [6]. Fewer farmers in the wet upper mid-altitude and highland MMEs correctly identified FAW than farmers in other agro-ecological zones. These two agro-ecological zones contribute more than 80% of the country's maize production, suggesting that the extension system may need additional capacity-building activities for farmers.

### FAW control strategies used by farmers

It seems the effectiveness of FAW control strategies varied by agro-ecology (Fig 3). Farmers in the wet lower mid-latitude and dry lowland agro-ecological zones named a few control methods. We asked farmers to score their effectiveness on a scale from zero (none) to ten (maximum). Insecticides received an average score of six (Fig 3 Panel A). The effectiveness of

**Table 3. Farmers' awareness and knowledge of FAW in the study areas (%).**

| Agro-ecological zones | Awareness of FAW (%) | Correctly identified FAW (%) |
|---|---|---|
| Wet upper mid-altitudes | 92.00 | 70.00 |
| Wet lower mid-altitudes | 100.00 | 100.00 |
| Dry mid-altitudes | 99.00 | 88.00 |
| Dry lowlands | 100.00 | 100.00 |
| Highlands | 92.00 | 80.00 |
| Average | 97.00 | 88.00 |

Source: Community survey.

chemicals remained the highest (Fig 3, Panels B-F). Cultural (e.g., rotation and fallow) and biological (e.g., caring for the striped earwig species during field management) pest control techniques received a score of five. The effectiveness of botanical extracts (e.g., neem-based products) and mechanical control (e.g., killing larvae of the pest) received below-average scores. The FGD participants gave a low effectiveness score (two) for agro-ecological approaches (e.g., cropping systems such as intercropping), which were promoted to control FAW in Ethiopia and elsewhere [2,7,18,56]. This result is in line with a study in southern Ethiopia that found that intercropping (maize-legume) had little impact on controlling maize production losses due to FAW in southern Ethiopia [10]. However, an experimental study in Uganda showed that intercropping was more effective than monocropping in controlling FAW and stemborers [57].

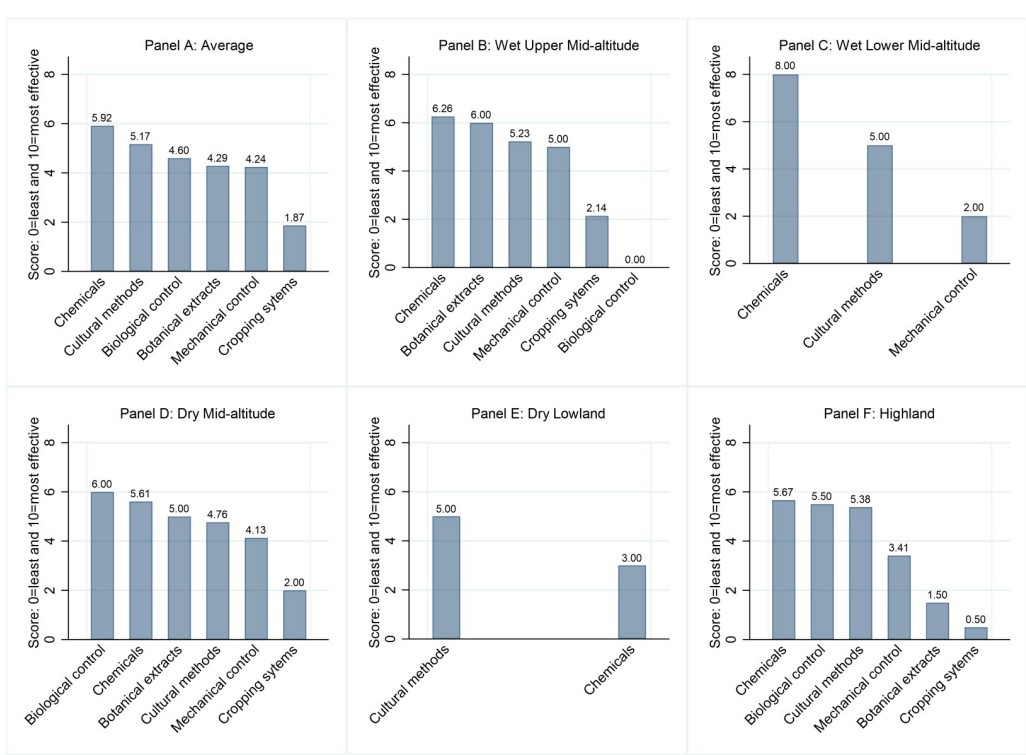

**Fig 3. Farmers' FAW control strategies by agro-ecological zones.**

## External support for FAW control

The impact of the pest may depend on the type of support the community receives to manage the pest. We asked the FGD participants to assess if the community they belonged to had received external support (e.g., training and funding) for controlling FAW. We also asked whether the support had increased, decreased, or remained the same. The support included training in FAW management, provision of credit, free insecticides, and spraying equipment. Farmers receive support from the regional and federal governments, Agricultural Research Systems, and development organizations. More than half of the communities (61%) had not received any support (Table 4). For 6% of the communities, support had remained the same, suggesting that farmers had received continuous support since the first occurrence of FAW in their respective communities. About 21% of the communities reported that external support for FAW control had increased. On the other hand, 11% of the studied communities reported that they had received external support, but it had decreased over time. The absence or low level of support may have contributed to higher production losses.

## Farmers affected by fall armyworm

The map in Fig 4 shows the distribution of the affected farmers by agro-ecology. The FGD results indicated that FAW affected 40% of maize farmers (Table 5), while the expert opinion interviews estimated that 51% of the farmers were affected (Table A1 in S3 File). In the wet lower mid-altitudes containing 3% of all maize farmers (Table A2 in S3 File), farmers were the most affected (59%). In the dry lowlands, where 4% of maize farmers are located, FAW affected 17% of farmers. For the other agro-ecological zones, the proportion of farmers affected by FAW was close to the country's average at 40%. The total number of farmers affected by FAW over the study period was 3.7 million per annum.

## Maize yield losses due to FAW

We estimated that, on average, FAW causes an annual loss of 36% in maize production (Table 6), despite the use of control measures. This estimate was close to the agricultural experts' estimate of 32% (Table A1 in S3 File). The map in Fig 5 shows the distribution of the yield losses by agro-ecology. In the wet mid-altitudes and highland agro-ecological zones, losses were close to the country's average of 36%. However, yield losses in the dry lowlands were higher than the country's average. This was perhaps because of the limited support farmers had received for FAW control in this zone (Table 4), the resulting limited use of control strategies (Fig 3), and the absence of hosts other than maize. The variability in yield loss could be due to several factors, including farming practices, natural enemies' availability, and climatic factors [18]. Several studies have established the role of climatic factors in FAW

**Table 4. FAW control support to the study communities.**

| External support: | Wet Upper Mid-altitudes | Wet Lower Mid-altitudes | Dry Mid-altitudes | Dry Lowlands | Highlands | Average |
|---|---|---|---|---|---|---|
| Not at all | 61.00 | 100.00 | 63.00 | 0.00 | 61.00 | 61.00 |
| Increased | 21.00 | 0.00 | 30.00 | 0.00 | 18.00 | 21.00 |
| Same | 7.00 | 0.00 | 0.00 | 100.00 | 5.00 | 6.00 |
| Decreased | 10.00 | 0.00 | 5.00 | 0.00 | 15.00 | 11.00 |
| Do not know | 2.00 | 0.00 | 1.00 | 0.00 | 0.00 | 1.00 |
| Total | 100.00 | 100 | 100.00 | 100.00 | 100.00 | 100.00 |

Source: Community survey.

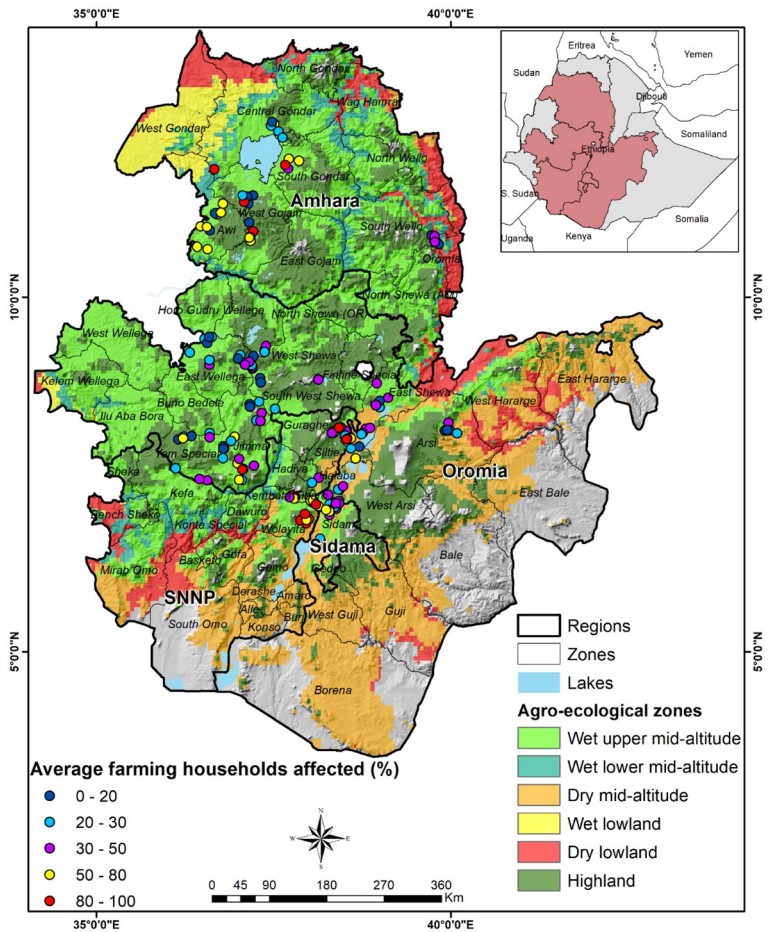

**Fig 4. Geographic distribution of average farmers affected (%) by FAW (2017–2019).**

**Table 5. Proportion of farmers affected by FAW (%).**

| Agro-ecological zones | 2017 | 2018 | 2019 | Average |
|---|---|---|---|---|
| Wet upper mid-altitudes | 36.40 | 39.64 | 39.71 | 38.60 |
| | (2.51) | (2.49) | (2.28) | (1.40) |
| Wet lower mid-altitudes | 55.00 | 55.00 | 67.50 | 59.17 |
| | (20.00) | (25.00) | (27.50) | (11.21) |
| Dry mid-altitudes | 44.02 | 41.71 | 45.27 | 43.68 |
| | (4.73) | (5.07) | (4.96) | (2.82) |
| Dry lowlands | 20.00 | 12.50 | 17.50 | 16.67 |
| | (0.00) [a] | (2.50) | (2.50) | (1.67) |
| Highlands | 37.95 | 42.39 | 44.75 | 41.67 |
| | (3.58) | (4.06) | (4.07) | (2.25) |
| Average | 38.07 | 40.68 | 42.13 | 40.30 |
| | (1.86) | (1.97) | (1.89) | (1.10) |

Note: Standard errors of the mean are reported in parenthesis;

[a] the standard errors are zero because FGD participants provided 20% loss for all data points.

Source: Community survey.

**Table 6. Yield losses due to FAW (%).**

| Agro-ecological zones | 2017 | 2018 | 2019 | Average |
|---|---|---|---|---|
| Wet upper mid-altitudes | 34.37 | 34.71 | 35.82 | 34.97 |
| | (2.14) | (1.96) | (2.05) | (1.18) |
| Wet lower mid-altitudes | 35.00 | 32.50 | 35.00 | 34.17 |
| | (5.00) | (2.50) | (5.00) | (2.01) |
| Dry mid-altitudes | 38.78 | 41.21 | 43.46 | 41.17 |
| | (3.78) | (3.47) | (3.15) | (1.99) |
| Dry lowlands | 80.00 | 80.00 | 80.00 | 80.00 |
| | (0.00) [a] | (0.00) [a] | (0.00) [a] | (0.00) [a] |
| Highlands | 33.20 | 36.43 | 34.20 | 34.61 |
| | (3.13) | (2.69) | (2.75) | (1.65) |
| Average | 35.13 | 36.64 | 36.96 | 36.25 |
| | (1.61) | (1.45) | (1.48) | (0.87) |

Note: Standard errors in parenthesis;

[a] the standard errors are zero because FGD participants provided 80% loss for all data points.

Source: Community survey.

incidence. The combined effect of natural enemies, including predators and parasitoids, could be up to 60% effective in controlling FAW if these natural enemies were conserved [29,58]. Heavy downpours can reduce FAW by washing away neonates and affecting the flight capability of adult moths. Soil health in terms of soil moisture and fertility enhance plant vigor, which, in turn, protects crops against heavy damage [59,60]. The yield loss estimates are lower than those reported in Kenya [6]. However, this comparison should be interpreted with caution as yield losses depend on several factors (agro-ecology, farm management, years of data collection, estimation approach, etc.).

## Production losses: Economic and food security implications

This sub-section reports the total maize production losses computed using Eq (1), presented by agro-ecological zones (Table 7) and administrative regions (Table A4 in S3 File). For 2017, we estimated Ethiopia lost 0.18 million tonnes of maize to FAW (Table 7). The production loss increased from 0.22 million tonnes in 2018 to 0.25 million tonnes in 2019. The increase in loss over time could be attributable to changes in the proportion of farmers affected (Table 5), the percentage yield losses (Table 6), the number of maize farmers (Table A2 in S3 File), and maize land size (Table A3 in S3 File). The highest production losses are in the wet upper mid-altitude, highland, and dry mid-altitude agro-ecological zones. The production losses are small compared to the first estimates by Day and colleagues [12,26]. For 2017, our estimate was 7% of the 2.74 million tonnes of maize production loss in Ethiopia, estimated by Day and colleagues [26]. Similarly, our estimated losses were 13% of the 1.67 million tonnes of maize loss in 2018 estimated by Rwomushana and colleagues [12].

Over the study period, the total production loss was 0.67 million tonnes of maize, generating an economic loss of US$ 200 million to the Ethiopian economy (Table 7). The economic loss is equivalent to 0.08% of the country's Gross Domestic Product (US$ 262 billion) from 2017 to 2019 [61]. Alternatively, the losses were equivalent to 3% of the total foreign direct investment (US$ 7,327 million) in 2017 and 2018 alone [4]. Using the 152 kg per capita consumption of maize in Ethiopia [62], the quantity of maize lost could have met the per capita

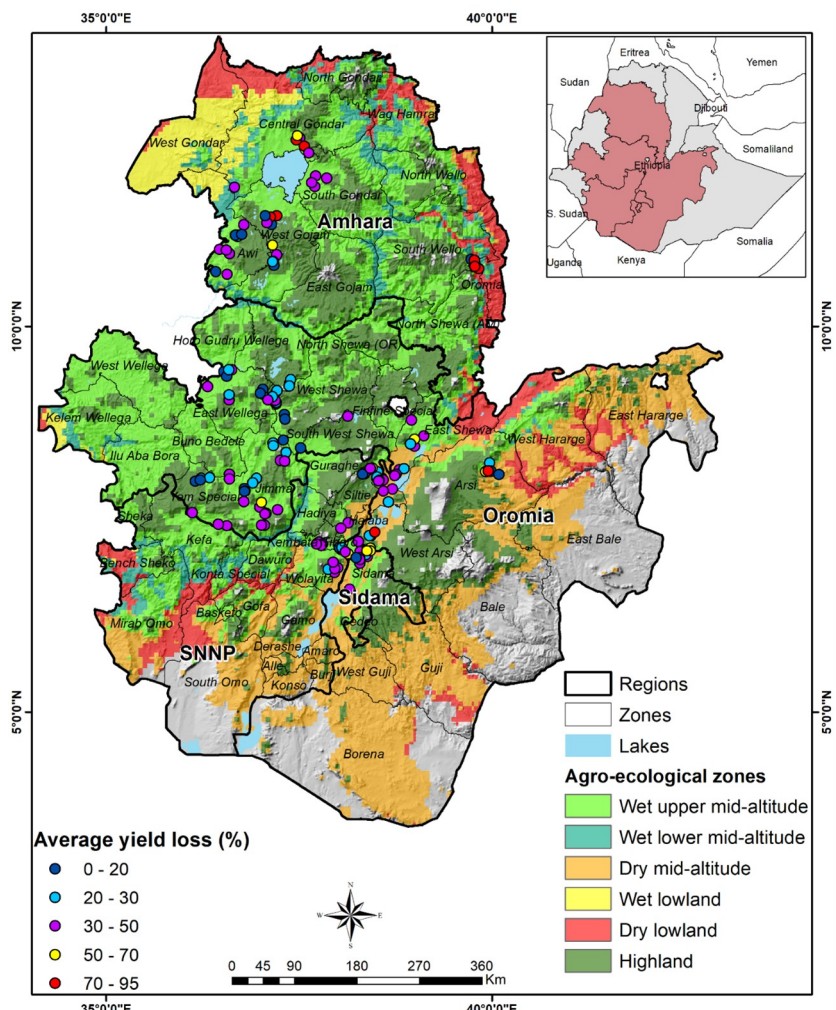

**Fig 5. Geographic distribution of average yield loss (%) due to FAW (2017–2019).**

**Table 7. Estimated total maize production losses.**

| MMEs | Loss (millions of tonnes) | | | Loss (millions of US$) ¥ | | |
|---|---|---|---|---|---|---|
| | **2017** | **2018** | **2019** | **2017** | **2018** | **2019** |
| Wet upper mid-altitudes | 0.064 | 0.080 | 0.084 | 15.21 | 22.35 | 28.53 |
| Wet lower mid-altitudes | 0.007 | 0.004 | 0.006 | 1.52 | 0.89 | 1.73 |
| Dry mid-altitudes | 0.049 | 0.047 | 0.073 | 13.33 | 14.94 | 24.87 |
| Wet lowlands | 0.002 | 0.005 | 0.005 | 0.55 | 1.37 | 1.53 |
| Dry lowlands | 0.002 | 0.002 | 0.002 | 0.75 | 0.58 | 0.46 |
| Highlands | 0.057 | 0.089 | 0.095 | 14.22 | 25.24 | 32.28 |
| Total | 0.182 | 0.228 | 0.265 | 45.59 | 65.38 | 89.40 |

¥ We use producer prices to estimate the value of production losses.

The exchange rate was 26.87 ETB/US$ in 2017, 27.43 ETB/US$ in 2018, and 29.23 ETB/US$ in 2019.

Source: Authors' computation based on community survey and CSA's agricultural sample survey.

maize consumption of over 50% (4.3 million) of the country's chronically food-insecure (8.5 million people) [63].

Although the economic and food security costs are high at the national level, the food security impact does not seem to be high at the household level-the per capita maize production loss is 60 kg per year (0.22 million tonnes divided by 3.7 million affected framers). A study in southern Ethiopia finds no significant effect of FAW on per capita maize consumption at the household level [10]. However, if the pest persists, it can have food security and poverty implication at the household level by reducing marketed surplus and income [10].

### Human and environmental effects of insecticides used for FAW control

Four insecticides were used in Ethiopia to reduce the impact of the pest (Table 8). According to the World Health Organization (WHO), malathion is slightly hazardous while the other chemicals are moderately hazardous [64]. All these insecticides have a high toxicity impact on the environment (e.g., by killing beneficial insects). Malathion, diazinon, and dimethoate carry considerable risk for the environment [51], as shown by the high EIQ values (Table 8). Synthetic insecticides are important management options for FAW control, but repeated application increases the accumulation of insecticides in the environment and raises major concern, as demonstrated by the high EIQ values. Furthermore, resistance to major classes of synthetic insecticides in the native regions of this pest is another problem. The efficacy of a synthetic insecticide-based management strategy is not guaranteed, as FAW has developed resistance to many active ingredients from different classes of insecticides [8,65–67]. This suggests an urgent need for resistance management as a vital component of integrated pest management. The risk impact on human health is relatively low, given the relatively low value of EIQ for consumers and producers. However, repeated exposure to small doses of insecticides can lead to long-term effects in humans. This calls for judicious and appropriate use of synthetic insecticides to successfully manage FAW and sustain the increased productivity of maize in Ethiopia and elsewhere in Africa. Previous reports show that Ethiopia is home to many natural enemies of FAW [68]. The adverse impacts of these insecticides on non-target and beneficial organisms and the environment might also explain pest incidence variations and yield losses because of the negative impact of insecticides on biological control agents. Our results suggest the importance of control strategies that effectively suppress the pest without compromising the natural environment. These may include biopesticides [69], predators, parasitoids, and pathogens [68,70], and push-pull strategies [18,71].

### Conclusions

Data on production losses are crucial for informed management of pests and evaluating the effectiveness of pest control measures. In this paper, we present the first comprehensive

**Table 8. Human health and environmental impacts of insecticides used to control FAW.**

| Insecticides | Active ingredient (%) | Application rate (liter/ha) | Quantity (liters) | Components of field use EIQ | | | |
|---|---|---|---|---|---|---|---|
| | | | | Average EIQ | Consumer effects | Producer effects | Ecological effects |
| Malathion | 50 | 2 | 114,529 | 23.80 | 3.80 | 7.70 | 49.60 |
| Diazinon | 60 | 1 | 256,914 | 22.60 | 1.30 | 3.50 | 63.00 |
| Dimethoate | 40 | 1 | 25,488 | 11.50 | 3.90 | 3.50 | 26.90 |
| Chlorpyrifos | 48 | 0.5 | 60,496 | 5.50 | 0.40 | 1.20 | 14.90 |

Source: Authors' computation based on MoA's pesticides data [37].

estimate of the impact of FAW on maize production, food security, and health in Ethiopia, contributing to the few existing studies in SSA. We used primary community survey data combined with a nationally representative agricultural household survey to achieve our objectives.

The pest caused significant economic and food losses at the country level. About 0.67 million tonnes of maize were lost, equivalent to 2.54% of the maize production (25.96 million tonnes) over the study period. This generates US$ 200 million (0.08% of the country's GDP) economic loss. These results vary by agro-ecology, which is vital for prioritizing investment. At the country's current 152 kg per capita consumption of maize, the maize lost to FAW could have met the maize consumption requirement of over 4 million food-insecure people. In the long run, together with other co-existing production constraints, FAW can put the livelihoods of many poor people at risk and may reverse the gains already made in productivity and poverty reduction that the country has achieved over the last three decades. Apart from the direct economic and food security losses, controlling FAW using pesticides contributes to environmental damage, threatening sustainable food production.

A key implication of these findings is that developing and promoting affordable, accessible, ecologically friendly control strategies must be facilitated to control the pest sustainably and effectively. Our analysis has the following caveats. First, we did not capture the total management costs, such as insecticides and labor costs, in controlling the pest. Second, although we indicate the toxicity of insecticides for the environment and human health, the insecticides application's health, and environmental costs (monetary costs to the society) are not factored into the analysis. Third, we cannot entirely rule out recall bias, although we recorded the yield and yield losses data agreed by each member of the FGD that can help minimize recall bias and avoid too high or too low estimates. We recommend future studies to (1) consider both the direct and total indirect costs of the pest to reflect its overall cost, and (2) introduce effective, healthy, and environmentally friendly management strategies for FAW control and conduct comprehensive agroecological zone-specific evaluations of their effectiveness.

## Supporting information

**S1 File. Data processing approach: Stata do file.**
(PDF)

**S2 File. The community survey data: Stata dataset file.**
(ZIP)

**S3 File. Appendix Tables A1-A4.**
(DOCX)

## Acknowledgments

We thank Zebdewos Selato and Hulubanchi Abera of the Ministry of Agriculture for their support and data on insecticides for FAW control in Ethiopia. We also thank Solomon Balew for supporting the research by designing the CSPro program during data collection, the enumerators, and supervisors for their dedication in conducting the survey, and the farmers and experts who participated in the study. We thank Rahel Solomon for processing the letters and bringing the Ethiopia's Central Statistical Agency datasets. Finally, we thank the anonymous reviewers for this insightful feedback that further improves the paper's quality.

## Author Contributions

**Conceptualization:** Zewdu Abro, Menale Kassie.

**Data curation:** Emily Kimathi.

**Formal analysis:** Zewdu Abro.

**Funding acquisition:** Tadele Tefera, Subramanian Sevgan, Menale Kassie.

**Investigation:** Zewdu Abro.

**Methodology:** Zewdu Abro.

**Project administration:** Tadele Tefera, Subramanian Sevgan.

**Resources:** Emily Kimathi.

**Software:** Zewdu Abro.

**Supervision:** Menale Kassie.

**Validation:** Emily Kimathi, Menale Kassie.

**Visualization:** Emily Kimathi, Hugo De Groote, Tadele Tefera, Subramanian Sevgan, Saliou Niassy, Menale Kassie.

**Writing – original draft:** Zewdu Abro.

**Writing – review & editing:** Emily Kimathi, Hugo De Groote, Tadele Tefera, Subramanian Sevgan, Saliou Niassy, Menale Kassie.

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
