## [Decision Letter · Decision Letter 0]

9 Jun 2021

PONE-D-21-14461

The socioeconomic and health impacts of Fall Armyworm in Ethiopia

PLOS ONE

Dear Dr. Kassie,

Thank you for submitting your manuscript to PLOS ONE. After careful consideration, we feel that it has merit but does not fully meet PLOS ONE’s publication criteria as it currently stands. Therefore, we invite you to submit a revised version of the manuscript that addresses the points raised during the review process.

Please address all comments by the reviewers.

We look forward to receiving your revised manuscript.

Kind regards,

Rodney N Nagoshi, Ph.D.

Academic Editor

PLOS ONE

4. We note that Figure 1, Figure 4 and Figure 5  in your submission contain map images which may be copyrighted. All PLOS content is published under the Creative Commons Attribution License (CC BY 4.0), which means that the manuscript, images, and Supporting Information files will be freely available online, and any third party is permitted to access, download, copy, distribute, and use these materials in any way, even commercially, with proper attribution. For these reasons, we cannot publish previously copyrighted maps or satellite images created using proprietary data, such as Google software (Google Maps, Street View, and Earth). For more information, see our copyright guidelines: http://journals.plos.org/plosone/s/licenses-and-copyright.

a. You may seek permission from the original copyright holder of Figure(s) [#] to publish the content specifically under the CC BY 4.0 license. 

5. We note that Figure 2 in your submission contain copyrighted images. All PLOS content is published under the Creative Commons Attribution License (CC BY 4.0), which means that the manuscript, images, and Supporting Information files will be freely available online, and any third party is permitted to access, download, copy, distribute, and use these materials in any way, even commercially, with proper attribution. For more information, see our copyright guidelines: http://journals.plos.org/plosone/s/licenses-and-copyright.

a. You may seek permission from the original copyright holder of Figure(s) [#] to publish the content specifically under the CC BY 4.0 license.

Reviewers' comments:

Reviewer's Responses to Questions

**Comments to the Author**

1. Is the manuscript technically sound, and do the data support the conclusions?

Reviewer #1: Yes

Reviewer #2: Partly

2. Has the statistical analysis been performed appropriately and rigorously? 

Reviewer #1: Yes

Reviewer #2: Yes

3. Have the authors made all data underlying the findings in their manuscript fully available?

Reviewer #1: Yes

Reviewer #2: Yes

4. Is the manuscript presented in an intelligible fashion and written in standard English?

Reviewer #1: Yes

Reviewer #2: Yes

5. Review Comments to the Author

Reviewer #1: Introduction, hypothesis, objectives, methodology, results and conclusions are clearly explained. References section needs revision based on the journal's guidelines. Also, some sections need to be moved to introduction and conclusion sections.

My comments and suggestions are provided in the attached file.

Reviewer #2: The study purported to investigate the socio-economic, environmental and health impacts of fall armyworm, an invasive pest that is causing devastating effects on maize production in Africa. The authors used data from focus group discussions (FGDs), combined with secondary household survey data covering three production seasons (2017-2019) in Ethiopia. They found that FAW infestation leads to average maize yield loss of 36%, as well as negative effects on the environment than on human health. Below are a few comments that could be considered when improving the paper.

1. It is questionable if the FGD participants were able to remember the actual and attainable yields in 2017 and 2018 when the FGDs were conducted in June-July 2020. Moreover, in lines 202-204, the authors wrote “Secondly, we asked them to estimate the attainable yield in the absence of production constraints. Thirdly, we asked farmers to quantify FAW’s contribution and other production constraints to the yield gap….”. Are you really sure that the farmers could reliably indicate how much yield they would have obtained in the absence of production constraints and the contribution of FAW? Thus, the yield loss estimates of 36% or 0.67 million tonnes of maize and the total economic loss to Ethiopia are based on data that are likely to be fraught with recall bias and measurement errors, raising questions about their reliability. The authors need to reflect on these concerns. I would have thought that a more reliable method would have been to compare the yield estimates of FAW-affected and non-affected households, as was done by a previous related study in Ethiopia, i.e., Kassie et al. 2020.

2. The authors found that while 97% of the FGD participants were aware of FAW, only 88% were able to correctly identify FAW. One may wonder how a farmer can be aware of FAW when they cannot identify it? In this context, it would be helpful to explain what is considered to be awareness of FAW. Does awareness imply that a farmer has at least heard about FAW?

3. In lines 248-250, the authors stated “Moreover, 88% of the 249 farmers in the FGDs correctly identified the FAW from the pictures shown (Table 3), slightly more than those in Kenya (82%) (De Groote et al., 2020).” Out of curiosity, I went to look at the De Groote et al. (2020) paper and found that the four pictures of insect pests labelled A1, A2, B and C in your manuscript are exactly as those presented in the De Groote et al. (2020) paper. Surprisingly, you did not attribute the source of the pictures to De Groote et al. (2020) or indicate the original source, if any. Wouldn’t this give rise to an issue of copyright infringement?

4. In lines 297-298, the authors wrote “…while the expert opinion interviews estimated that 51% of the farmers were affected (Table A1, Appendix)”. This percentage and others in Table A1 seem to be unreliable and based on expert guesstimates because there is no indication that the 180 experts sought the opinions of farmers on FAW or observed farmers’ fields for FAW attack. How did the experts know how many farmers were affected by FAW? Same applies to the expert estimates of yield loss due to FAW.

5. Finally, I have reservations that you were actually able to estimate the effect of FAW on human health and the environment, based on the data used. For example, it is indicated in the abstract that “We also find that the application of insecticides to control FAW has more significant toxic effects on the environment than on humans”. However, you did not provide any evidence on insecticides used by the sampled farmers for FAW control. Your EIQ estimations were based on MoA’s pesticides data (government distributed pesticides) and application rates that may not reflect the farmers’ actual pesticide application practices against FAW. Moreover, the use of these pesticides may not have an effect on human health if the farmers use the appropriate personal protective equipment and observe the necessary safe pesticide practices, and vice versa. It appears your results may rather be reflective of the potential health and environmental effects of the listed pesticides in Ethiopia, regardless of whether or not they are used for FAW control.

6. PLOS authors have the option to publish the peer review history of their article (what does this mean?). If published, this will include your full peer review and any attached files.

Reviewer #1: No

Reviewer #2: No

---

## [Author Response · Author response to Decision Letter 0]

1 Sep 2021

Responses to the two reviewers and the Academic Editor feedback uploaded or attached along with other files.

---

## [Editor Report · Decision Letter 1]

9 Sep 2021

The socioeconomic and health impacts of Fall Armyworm in Ethiopia

PONE-D-21-14461R1

Dear Dr. Kassie,

We’re pleased to inform you that your manuscript has been judged scientifically suitable for publication and will be formally accepted for publication once it meets all outstanding technical requirements.

Kind regards,

Rodney N. Nagoshi, Ph.D.

Academic Editor

PLOS ONE
---

## [Editor Report · Acceptance letter]

28 Oct 2021

PONE-D-21-14461R1 

Socioeconomic and health impacts of Fall Armyworm in Ethiopia 

Dear Dr. Kassie:

I'm pleased to inform you that your manuscript has been deemed suitable for publication in PLOS ONE. Congratulations! Your manuscript is now with our production department. 

Kind regards, 

on behalf of

Dr. Rodney N. Nagoshi 

Academic Editor

PLOS ONE